# Long-Term Outcomes in Patients on Life-Long Antibiotics: A Five-Year Cohort Study

**DOI:** 10.3390/antibiotics11010062

**Published:** 2022-01-05

**Authors:** Christopher Kiss, Declan Connoley, Kathryn Connelly, Kylie Horne, Tony Korman, Ian Woolley, Jillian S. Y. Lau

**Affiliations:** 1Monash Infectious Diseases, Monash Medical Centre, Monash Health, Clayton, VIC 3168, Australia; c.r.kiss89@gmail.com (C.K.); declan.connoley@gmail.com (D.C.); kathryn.connelly@monash.edu (K.C.); Kylie.Horne@monashhealth.org (K.H.); tony.korman@monash.edu (T.K.); ian.woolley@monash.edu (I.W.); 2Faculty of Medicine, Nursing and Health Sciences, Monash University, Clayton, VIC 3800, Australia

**Keywords:** antibiotics, life-long, suppression, antimicrobial resistance, multi-resistant organisms

## Abstract

Background: Little is known about the impacts at an individual level of long-term antibiotic consumption. We explored health outcomes of long-term antibiotic therapy prescribed to a cohort of patients to suppress infections deemed incurable. Methods: We conducted a 5-year longitudinal study of patients on long-term antibiotics at Monash Health, a metropolitan tertiary-level hospital network in Australia. Adults prescribed antibiotics for >12 months to suppress chronic infection or prevent recurrent infection were included. A retrospective review of medical records and a descriptive analysis was conducted. Results: Twenty-seven patients were followed up during the study period, from 29 patients originally identified in Monash Health in 2014. Seven of the 27 patients (26%) died from causes unrelated to the suppressed infection, six (22%) ceased long-term antibiotic therapy and two (7%) required treatment modification. Fifteen (56%) were colonised with multiresistant microorganisms, including vancomycin resistant Enterococci, methicillin resistant *Staphylococcus aureus*, and carbapenem resistant Enterobacteriaciae. Conclusions: This work highlights the potential pitfalls of long-term antibiotic therapy, and the frailty of this cohort, who are often ineligible for definitive curative therapy.

## 1. Introduction

Antibiotics were originally developed and then used as short-term therapy for bacterial infections [1]. Evaluation of their utility was thus focused on short term harm and benefits. Subsequent evidence supports specific long-term effective use of certain antibiotics for prophylaxis against infection following transplantation, splenectomy and for recurrent urinary tract infections (UTI) [2], though less evidence supports their use in other settings, such as for infected prosthetic material not amenable to removal and for nonantibacterial effects such as immunomodulation [2]. In this context, the evidence is limited, as is the measurement of the adverse consequences of long-term therapy [3].

Pitfalls associated with long-term antibiotic use include development of antimicrobial resistance (AMR) and cumulative risk of adverse events. AMR is a growing global concern with multiresistant microorganisms (MROs) responsible for greater than 35,000 deaths annually in the United States [4]. Emergence of AMR has been demonstrated after prolonged antibiotic usage in multiple conditions including post-splenectomy, rheumatic fever and UTI prophylaxis [2].

We previously reviewed antibiotic prescribing practices at Monash Health, a large tertiary-level, university-affiliated health service in Melbourne, Australia, and found that long-term antibiotic therapy made up only a small proportion (202/66,127, 0.3%) of total prescriptions, with great heterogeneity in indication for use [5]. Long-term therapy was defined as an intended treatment course of greater than 12 months. Out of the 202 patients on long-term antibiotic therapy identified in this previous study, 29 patients were prescribed this therapy for infections deemed incurable [3,5]. We looked more closely at this cohort of 29 patients and found one in five patients screened to be colonised with multiresistant microorganisms (MROs) including vancomycin resistant Enterococci (VRE) and methicillin resistant *Staphylococcus aureus* (MRSA) [3]. Furthermore, nearly half of the patients in this cohort described adverse drug reactions (ADRs) attributable to long-term antibiotic consumption [3].

Despite being a recognised treatment strategy [2], there is limited information on the outcomes in patients who are prescribed long-term antibiotics for suppression of infections deemed otherwise uncurable. The aim of this study was to describe the impact of long-term antibiotic use on health outcomes in this cohort of patients originally identified in 2015, and explore issues such as tolerance of therapy, treatment modification, adverse events and colonisation with MROs.

## 2. Results

### 2.1. Patient Demographics

Of the 29 patients originally identified in our previous study [3], two had no follow-up through our institution and were excluded from this analysis. Baseline demographics and clinical information for the 27 eligible patients are presented in Table 1. The median age was 73 (range of 47–93). Fourteen (52%) were female. Fifteen different antibiotic regimens were used, with combination rifampicin and fusidic acid (6/27, 22%), and cefalexin monotherapy (6/27, 22%) prescribed most commonly. Indications for long-term antibiotic prescriptions were diverse, the most frequent being prosthetic joint infection (PJI) (15/27, 56%), vascular graft infection (VGI) (3/27, 11%), and recurrent *Staphylococcus aureus* bacteraemia (3/27, 11%). The Charlson Comorbidity Index (CCI) increased over the study period from 5 (median, range 1–13, interquartile range 3–7) at baseline, to 7 (median, range 1–13, interquartile range of 4–9) by the end of follow-up.

### 2.2. Patient Outcomes

Patient outcomes are summarised in Figure 1. At the conclusion of the study period seven patients had died (26%), two of whom died from infection, (not the original suppressed infection). No deaths were attributed to treatment-related adverse drug reactions. Six patients (22%) ceased long-term antibiotic therapy: three completed their therapy, two ceased because of adverse drug reactions including acute kidney injury (AKI) and vaginal thrush, and one patient ceased therapy due to anticipated drug interactions with new medications. These six patients had previously taken prolonged antibiotics for between 2 to 10 years. They were followed up for between 3 months and 2 years following cessation of antibiotics prior to discharge from clinic. Four had prosthetic joint infections and had further surgery (two had joint revision surgery, one had washout/liner exchange, one had joint arthroscopy without washout). All six were discharged from infectious diseases (ID) outpatient follow-up and have not had any relapse of their suppressed infection.

### 2.3. Follow Up

Of the 14 patients who had remained on long-term antibiotic therapy (52%), eight had been discharged from ID without any plan to cease therapy, tolerating their regimen without issue. Three of these eight were discharged with no further input from our healthcare network. The other five were followed up by specialty units within the network including orthopaedics, cardiology and haematology, and are clinically stable. Of the six patients with ongoing follow-up, three are reviewed annually, two reviewed six-monthly, and one every three months. Two patients had modifications to their regimen, one because of diarrhea, and another because a new organism was identified in a subsequent surgical procedure. Four patients remained on their initial regimen and continue to attend ID outpatients.

### 2.4. Adverse Events

Twenty patients required hospital admission in the follow-up period, 15 of these for an infection unrelated to the one requiring suppressive therapy. Two patients experienced adverse drug events newly identified in the follow up period. One had severe cholestatic liver function derangement (peak GGT 1562, ALP 600) and significant kidney injury (eGFR 37 from 75) with associated metabolic acidosis, which occurred two years following commencement of suppressive therapy. The other patient experienced recurrent vaginal thrush.

The rest of the reported adverse drug events were noted during our original study [5], and included gastrointestinal side effects such as nausea, vomiting, stomach discomfort, reflux and diarrhoea, itch/rash, weight gain, thrush, confusion, alopecia, fatigue, and headache).

### 2.5. MRO Colonisation

We found 18 patients to be colonised with MROs (18/27, 67%). Of these patients, vancomycin resistant Enterococci (VRE) was isolated in eight patients, methicillin resistant *Staphylococcus aureus* (MRSA) in nine patients, carbapenem resistant Enterobacteriaciae (CRE) in one patient, and five patients had extended-spectrum beta-lactamase organisms (ESBL) isolated (three with *Escherichia coli*, two with *Klebsiella pneumoniae*). Six patients had more than one MRO identified. Of the 24 organisms isolated across the 18 patients, 14 organisms were isolated from clinical specimens, the remaining 10 organisms were isolated from screening purposes either from admission infection control or research purposes.

Eight patients had MROs detected after the commencement of suppressive therapy (ranging between 6 months to 5 years following commencement). Eight patients had MROs detected at the time of commencement of suppressive therapy, and two had MROs detected prior to suppressive therapy (both detected 1 year prior).

## 3. Discussion

We observed a cohort of patients who had been prescribed long-term antibiotics over a five-year period. The key finding of our study was a high overall mortality rate of 26% over five years. Twenty-two percent of the cohort ceased their antimicrobials, with intolerable medication side effects being the reason for cessation in a third of these cases.

The mortality rate of 26% is higher than in previous studies focusing on single indications for long-term antibiotic therapy. A mortality rate of 17.4% was previously demonstrated in a cohort of 23 patients with infected prosthetic hip joints after a mean follow up of 33 months [6]. Reassuringly, no patients in our study died from the underlying infection requiring suppressive therapy, or from adverse effects of the suppressive therapy itself. Factors such as older age and comorbidities likely contributed to the observed mortality rate. This is supported by the high median CCI of five of our cohort, which has an estimated 1-year mortality of 85% [7], as well as the admission frequency for both infective and noninfective indications seen over the follow-up period, unrelated to chronic infection and suppressive antibiotics. We postulate that patients selected for suppressive antibiotic therapy are likely ineligible for definitive (and usually more aggressive) management, reflecting a frailer cohort with more comorbidities. Such patients are also at higher risk of adverse effects from long-term antibiotics, which was seen in our study with 19% needing modification or cessation of their treatment regimen most commonly due to adverse drug reactions or medication interactions. This highlights the complexities of managing these patients over time, and balancing infection suppression with the risks of long-term antibiotics in the context of comorbidities and polypharmacy.

No recurrences of infection were noted in the patients who ceased antibiotic therapy and continued to be followed up in our healthcare network. This may indicate that these patients may not have actually required long-term antibiotics for their original infective indications. Alternatively, this may reflect the small sample size of our cohort, as it does not match previous studies that have stopped long-term antibiotics following prosthetic joint infection [8].

A major strength of this study is the longitudinal design with long-term follow-up over five years. There are few similar studies [2], of which most have focused on suppressive antibiotics in the setting of a particular intervention such as debridement and implant retention for PJI [6,9], or VGI [10]. These studies did not measure outcomes such as rates of continued follow-up in ID clinic, and antibiotic discontinuation.

Our study has several limitations. First, the small cohort size from a single healthcare network, and second reliance on medical records as the sole source of data. Nevertheless, this study is important because long-term antibiotic prescribing may contribute significantly to antibiotic consumption and associated morbidity, particularly in indications where the evidence for their use is unclear. MROs were still detected in 67% of our cohort from clinical specimens and screening for infection control and previous study purposes. This suggests an association between long-term antibiotic consumption and AMR, which is likely underestimated in our study as there was no routine MRO screening. Another limitation is the heterogeneity of our cohort, including differing indications and antibiotic regimens. Although this limits our ability to draw specific conclusions, it does reflect the real-life challenges of managing these heterogeneous and often medically complex patients on long-term antibiotics.

## 4. Materials and Methods

We conducted a longitudinal study of patients taking long-term antibiotics at Monash Health, a large tertiary-level, university affiliated health service in Melbourne, Australia comprising five hospitals and servicing more than 1.5 million people. Patients were followed up for five years between April 2015 and May 2020. This study was approved by the Monash Health Human Research Ethics Committee: approval number 14379A.

Adult patients prescribed long-term antibiotics were identified at baseline using the hospital drug management system (Merlin Ver. 4.94, Pharmhos Software, Port Melbourne, VIC, Australia) as previously described [3,5]. Patients prescribed antibiotics for longer than 12 months to suppress chronic infection or prevent recurrent infection were included. Patients were excluded if they were aged under 18 years, unable to give informed consent, if they were prescribed prophylaxis in the context of immunocompromise and if they received antibiotic therapy outside of our hospital network.

This study reports five-year follow-up of this patient cohort. Medical records were retrospectively reviewed to determine baseline demographic characteristics and outcomes, including rates of ongoing antibiotic therapy and cessation, adverse events (including side effects attributable to antibiotic therapy and infection-related hospital admission), mortality, and isolation of MROs from clinical specimens, infection control screening and our previous study’s swabs [3]. A CCI was calculated as a validated method to identify comorbidity extent in our population at baseline and the end of the follow-up period [11]. A descriptive analysis was performed.

## 5. Conclusions

Our study demonstrates the pitfalls of long-term antibiotic therapy. A high 5-year mortality rate reflects the older and more comorbid population of patients being prescribed this therapy. Further research is needed to elucidate which antimicrobials are associated with higher risk of adverse drug events and failure in this population, along with detailing true rates of MRO colonisation.

## Figures and Tables

**Figure 1 antibiotics-11-00062-f001:**
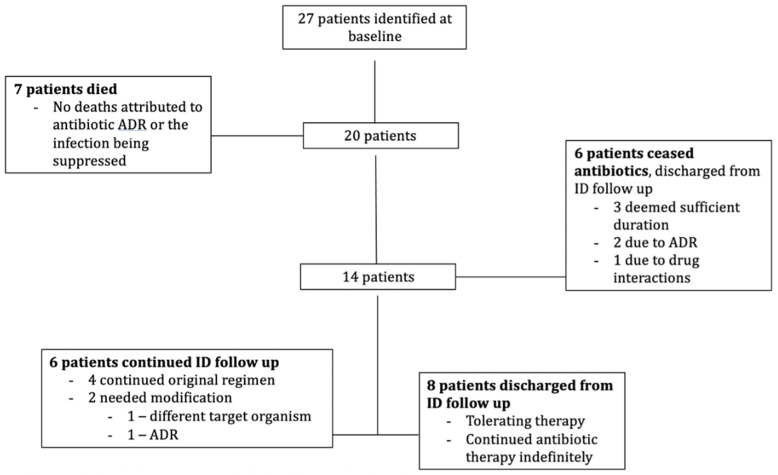
Patient outcomes. ADR, adverse drug reaction; ID, infectious diseases.

**Table 1 antibiotics-11-00062-t001:** Patient characteristics.

Age	Sex	Antibiotic Regimen	Indication	Targeted Organism/s	Charlson Comorbidity Index Baseline End		MRO—Isolate Site	Cause of Death	Hospital Admission Days (Total/Infection Related/ICU)	Ongoing ID Follow Up	Duration of Follow Up (Years)	Regimen Adjustment/Reason	Adverse Drug Reactions ^
70	M	Amoxycillin	PVI	*E. faecalis*	13	13	ESBL *E. coli*—Urine	Lung Cancer	13/7/0	N/A	5 months	N/A	
76	F	Rifampicin, Fusidic Acid, Pristinamycin	Chronic hip osteomyelitis (OM)	MRSA	3	7	MRSA—hip tissue VRE—Screening swab	Unknown (coroner’s case)	109/64/5	N/A	7	N/A	
79	M	amoxicillin/clavulanic acid, Ciprofloxacin	PJI	*S. marcescens*	11	12	VRE—Screening swab	Heart failure	64/30/0	N/A	4	Ceased amoxicillin/Deemed unnecessary + ADR	Oral Thrush
56	F	Flucloxacillin	PJI	MSSA	4	6	VRE—Screening swab ESBL *E. coli*—urine	Unknown (coroner’s case)	106/102/14	N/A	4	Added ciprofloxacin/Relapsed infection	Reflux
70	M	Cephalexin	Recurrent MSSA bacteraemia	MSSA	6	7	VRE—Screening swab	Chest sepsis	17/15/15	N/A	4	N/A	
87	F	Rifampicin, Fusidic Acid, ciprofloxacin	PJI	MRSA	10	10	MRSA—joint tissue	Myocardial Infarction	7/0/0	N/A	2	N/A	
63	F	Rifampicin, Amoxycillin, Doxycycline	PJI	*C. amycolatum*, *E. faecalis*, *S. epidermidis*	2	2	ESBL *E. coli*—urine	Intra-abdominal haemorrhage	22/22/2	N/A	2	N/A	
65	M	Cephalexin	Recurrent MSSA bacteraemia	MSSA	3	7	No	N/A	86/79/2	No	5	Ceased/deemed unnecessary	
65	M	Rifampicin, Fusidic Acid	PJI	MRSA	5	5	MRSA—joint tissue	N/A	9/8/0	No	5	Ceased/projected medication interactions	
68	M	Pristinamycin, Cotrimoxazole, Ciprofloxacin, Fluconazole	VGI	*S. maltophilia*, VRE, *P. moteilii*, *C. albicans*	4	7	VRE—abdominal pus ESBL *K. pneumoniae*—screening swab	N/A	4/4/0	No	4	Ceased/ADR	Acute kidney and liver injury
70	M	Amoxycillin	PJI	*S. agalactiae*	6	8	ESBL *E. coli*—urine	N/A	7/2/0	No	1	Ceased/Deemed unnecessary	
56	F	Cephalexin	PJI	MSSA	4	7	No	N/A	0/0/0	No	3	Ceased/Deemed unnecessary	Vaginal thrush
59	F	Cephalexin	PJI	*S. agalactiae*	6	6	MRSA—wound swab	N/A	41/36/0	No	4	Ceased/Inefficacy + ADR	Vaginal thrush
65	F	Flucloxacillin, Ciprofloxacin	PJI	MSSA, *C. aurimucosum*	5	5	VRE—screening swab	N/A	360/314/8	Yes	7	Changed to doxycycline/Failed definitive surgery, new target organism	
69	F	Rifampicin, Fusidic Acid	PJI	MRSA	3	4	MRSA—joint tissue	N/A	25/0/0	Yes	5	N/A	
74	M	amoxicillin-clavulanic acid Pristinamycin	VGI	*S. typhimurium*, VRE	6	10	VRE, CRE, *E. cloacae*—screening swab	N/A	3/0/0	Yes	5	Changed amoxicillin/clavulanic acid to amoxicillin/ADR	Diarrhoea
78	M	Rifampicin, Fusidic Acid	PJI	MRSA	5	13	MRSA, ESBL K. pneumoniae—Knee tissue	N/A	12/9/0	Yes	6	N/A	
45	F	Rifampicin Fusidic Acid	PJI	*S. epidermidis*, MSSA	1	2	No	N/A	30/30/0	Yes	6	N/A	
69	M	Rifampicin, Fusidic Acid	PJI	MRSA	2	3	MRSA—joint tissue	N/A	0/0/0	Yes	9	N/A	
77	M	Cephalexin	PJI	MSSA	4	5	No	N/A	0/0/0	No	5 months	N/A	
73	F	Amoxycillin	Infected spinal metalware	PSSA	9	9	No	N/A	0/0/0	No	2 months	N/A	
45	F	Penicillin	PVI	*C. acnes*	1	3	No	N/A	0/0/0	No	1	N/A	
42	F	Nitrofurantoin	Recurrent urinary tract infection	*E. faecalis*	1	1	No	N/A	29/0/0	No	1	N/A	
59	M	Clindamycin, Amoxycillin	VGI	*S. epidermidis*	3	4	No	N/A	0/0/0	No	1	N/A	
88	F	Cephalexin	Recurrent MSSA bacteraemia, OM	MSSA	7	7	MRSA, VRE—screening swab	N/A	0/0/0	No	1	N/A	
86	F	Rifampicin, Fusidic Acid	PJI	MRSA	9	9	MRSA—joint tissue	N/A	34/26/0	No	4	N/A	
71	M	Ciprofloxacin	Post laminectomy infection	*E. cloacae*, *P. mirabilis*	7	7	No	N/A	2/0/0	No	1	N/A	

M—male; F—female; PJI—prosthetic joint infection; PVI—prosthetic valve infection; MSSA—methicillin susceptible *Staphylococcus aureus*; OM—osteomyelitis; VGI—vascular graft infection; MRO—multi-resistant microorganisms; MRSA—methicillin resistant *Staphylococcus aureus*; VRE—vancomycin resistant Enterococci; CRE—carbapenem resistant Enterobacteriaciae; ICU—intensive care unit; ADR—adverse drug reaction. Charlson Comorbidity Index—baseline refers to time of recruitment to study in 2015, end refers to the last available review. MRO; multi-resistant organism; ^ reported during follow-up period.

## Data Availability

The data used in this study has been presented in table format in the results section. Any further deidentified information can be made available in deidentified format upon reasonable request.

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
