# Peer review of "Long-Term Outcomes in Patients on Life-Long Antibiotics: A Five-Year Cohort Study"

_antibiotics, 2022, doi:10.3390/antibiotics11010062_

Round 1

Reviewer 1 Report

Line 44: How were these 29 patients identified? Do you expect this represented all the patients on long-term therapy at your center, or were some patients potentially missed? 

   ---    Line 153: Seems that you used a system called Merlin, but it might be nice to include one extra line about how this works as I am not familiar with this system. Also, is Merlin a proprietary product? If so, should indicate the company and location. 

Table 1: It would be interesting if you could specify how long each patient had been followed at the time of your data collection or until the end points you identified. This is purely a format issue, but it is very difficult to read this table given the narrow columns cutting off most of the words. Consider using a smaller font if possible or else removing less important columns such as MRO isolate site, hospital admission days, and perhaps cause of death. 

Line 89: Could you specify if the 8 patients who were discharged had any additional follow-up data? I think it makes a difference in terms of outcomes if a patient was known to be discharged and continued to do well for several years versus discharged and had no further follow-up information. 

Line 97: I think you should elaborate on the two adverse drug events a little bit, given that this is one of your most important outcomes. Similarly, you may want to describe the one drug-drug interaction you observed. 

Line 98: I'm not sure why you are including MRO colonization in adverse events. I would expect adverse events only to include events attributed to be secondary to antibiotic use, whereas MRO seems to be a demographic characteristic. 

Line 117: On a related note, it is interesting that you observed no recurrences. Why do you think that was the case? Sample size alone? Perhaps you could provide a brief discussion of previously reported recurrence rates at least for PJI and provide any additional comments as this relates to your study. 

Line 141: You didn't really mention antimicrobial resistance emergence in your results as far as I can tell, thus, it was not so much that is was underestimated as it was data not available to you at all. This is significant given that you described AMR development as one of the main concerns with long-term antibiotic suppression. 

Supplementary file: This is currently identical to your Table 1. If you remove columns from table 1 such as I suggested above, then it perhaps makes sense to keep this. Otherwise, would omit. 

Very nice study. We commonly use this approach but most of the outcomes related studies that I have seen focus on rate of recurrence of infection. There are few real conclusions that you can draw from such a small sample size, however, I think others could use this study as a model for studies including larger cohorts and/or even longer term follow-up. 

Author Response

We are very grateful for the time you have taken to provide these insightful comments and recommendations. We have responded to each one as below, and referred to where changes have been made in the manuscript.

Line 44: How were these 29 patients identified? Do you expect this represented all the patients on long-term therapy at your center, or were some patients potentially missed? 

These 29 patients were identified from a previous study conducted at our network (Lau et. al. 2017, reference 5 in this manuscript). These 29 patients were part of a larger cohort of patients identified through interrogating pharmacy dispensing records at our healthcare network, and were prescribed antibiotics for greater than 12 months for an infection deemed incurable. The following sentences have been added into the introduction section to describe this cohort more thoroughly from line 46:

“Out of the 202 patients on long-term antibiotic therapy identified in this previous study, 29 patients were prescribed this therapy for infections deemed incurable (3, 5). We looked more closely at this cohort of 29 patients and found 1 in 5 patients screened to be colonised with multi-resistant microorganisms (MROs)…”

We believe that these 29 patients identified in the original study did represent an accurate snapshot of the number of patients on long-term antibiotics (as per our definition) at the time, as we reviewed all dispensed antibiotics at our healthcare network over a 6-month period. We may have missed a small number of patients who received their prescription from the hospital but then had their medications dispensed elsewhere, but we do not believe that this made a significant difference to our overall results. These limitations were discussed in the original publication.

   ---    Line 153: Seems that you used a system called Merlin, but it might be nice to include one extra line about how this works as I am not familiar with this system. Also, is Merlin a proprietary product? If so, should indicate the company and location. 

Once again this was described in the original paper (reference 5) but the following sentence was added to the manuscript from line 169:

“Eligible patients were identified using the hospital drug management system (Merlin Ver. 4.94, Pharmhos Software, Port Melbourne, Victoria, Australia) as previously described (3).”

Table 1: It would be interesting if you could specify how long each patient had been followed at the time of your data collection or until the end points you identified. This is purely a format issue, but it is very difficult to read this table given the narrow columns cutting off most of the words. Consider using a smaller font if possible or else removing less important columns such as MRO isolate site, hospital admission days, and perhaps cause of death. 

Thank you for this comment. We have added an extra column to table 1 to describe the duration of follow up for each patient. The table formatting has also been improved to allow all columns and words to be legible.

Line 89: Could you specify if the 8 patients who were discharged had any additional follow-up data? I think it makes a difference in terms of outcomes if a patient was known to be discharged and continued to do well for several years versus discharged and had no further follow-up information.

Thank you for this suggestion. We agree it is important to provide this additional information and have added the following sentences to reflect this from line 85:

“Three of these eight were discharged with no further input from our healthcare network. The other five were followed up by specialty units within the network including (orthopaedics, cardiology and haematology, and are clinically stable.” 

Line 97: I think you should elaborate on the two adverse drug events a little bit, given that this is one of your most important outcomes. Similarly, you may want to describe the one drug-drug interaction you observed. 

Thank you for this suggestion. We agree it is important to provide this additional information and have added the following sentences to reflect this from line 120:

One had severe cholestatic liver function derangement (peak GGT 1562, ALP 600) and significant kidney injury (eGFR 37 from 75) with associated metabolic acidosis, which occurred two years following commencement of suppressive therapy. The other patient experienced recurrent vaginal thrush.

Line 98: I'm not sure why you are including MRO colonization in adverse events. I would expect adverse events only to include events attributed to be secondary to antibiotic use, whereas MRO seems to be a demographic characteristic. 

Thank you for this comment. We agree that MRO colonisation should be in it’s own section and not lumped with adverse events. This is now section 2.5.

Line 117: On a related note, it is interesting that you observed no recurrences. Why do you think that was the case? Sample size alone? Perhaps you could provide a brief discussion of previously reported recurrence rates at least for PJI and provide any additional comments as this relates to your study. 

We agree that this is an interesting observation that should be noted and discussed. We have added the following text to address this from line 158.

“No recurrences of infection were noted in the patients who ceased antibiotic therapy and continued to be followed up in our healthcare network. This may indicate that these patients may not have actually required long-term antibiotics for their original infective indications. Alternatively, this may reflect the small sample size of our cohort, as it does not match previous studies that have stopped long-term antibiotics following prosthetic joint infection (9).”

Line 141: You didn't really mention antimicrobial resistance emergence in your results as far as I can tell, thus, it was not so much that is was underestimated as it was data not available to you at all. This is significant given that you described AMR development as one of the main concerns with long-term antibiotic suppression. 

We do not think it’s possible in this study for us to infer the emergence of AMR in this cohort due to their consumption of long-term antibiotics, as we do not have microbiology or screening results from pre-commencement of therapy. This is discussed in our limitations section and highlights the importance of closely studying and following up this cohort.  

Supplementary file: This is currently identical to your Table 1. If you remove columns from table 1 such as I suggested above, then it perhaps makes sense to keep this. Otherwise, would omit. 

Thank you, this was an error made when submitting the files. The supplementary file has been deleted.

Very nice study. We commonly use this approach but most of the outcomes related studies that I have seen focus on rate of recurrence of infection. There are few real conclusions that you can draw from such a small sample size, however, I think others could use this study as a model for studies including larger cohorts and/or even longer term follow-up. 

Thank you again for your insightful comments and recommendations. We agree that this work should be repeated in a much larger national cohort to shed further light on this not well studied topic of antibiotic prescribing, and have planned further projects modelled on this study. We hope that this revised manuscript is appropriate for publication in Antibiotics.

Reviewer 2 Report

The manuscript is interesting to read. I do not have major commets. In Table 1, PVI is not explained.

Author Response

Dear reviewer, 

Thank you for your time in reviewing our manuscript. Your comments are much appreciated. We have added "PVI" into the legend for table 1, which is an abbreviation for prosthetic valve infection.

Sincerely, 

Jillian Lau

Reviewer 3 Report

In their Brief Report, Kiss et al. describe 27 patients on long-term antibiotic therapy, defined as longer than 12 months. Most frequent reasons for antibiotic treatment were prosthetic joint infection and vascular graft infection. Follow-up was performed for five years. According to their data, no patient died due to the suppressed infection or due to side effects. However, intolerable side effects were the reason for cessation of treatment in about 7% of patients. 67% of patients were colonized with MROs.

I think it is an important report as long-term antibiotic treatment is unavoidable in some cases, but not much is known about development of bacterial resistance or side effects. The cohort is quite inhomogeneous and a more systematic analysis is not possible. However, it reflects certainly the real-life situation in an ID department.

Can the authors provide the timing of screening for MROs? Were the patients positive for MROs at the start of antibiotic treatment or did this develop while being given antibiotics? Did any of the patients develop an infection with the colonizing MRO in the observation period?

I also want to ask the authors to describe the side effects of the antibiotic therapy in more detail, including gastro-intestinal side effects (e.g. what about nausea, etc.). The side effects mentioned in the table 1: were these the reasons for the cessation of treatment?

What about the patients who stopped treatment and did not have any recurrence. Can the authors describe these patients in more detail, e.g. what was the time interval for stopping treatment, which diagnoses, was any additional treatment done (like surgery), … Can they give a hypothesis why the cessation of antibiotic treatment was possible? Is there a certain time period that is long enough?

Author Response

Dear reviewer,

We are very grateful for the time you have taken to provide these insightful comments and recommendations. We have responded to each one as below, and referred to where changes have been made in the manuscript.

Can the authors provide the timing of screening for MROs? Were the patients positive for MROs at the start of antibiotic treatment or did this develop while being given antibiotics? Did any of the patients develop an infection with the colonizing MRO in the observation period?

We thank you for this important question. Our study recorded microbiological data that was available from electronic medical records, and results represented both screening swabs and clinical samples. We have added the following text to further describe when MRO were identified in patients where this data was available from line 133.

“Eight patients had MROs detected after the commencement of suppressive therapy (ranging between 6 months to 5 years following commencement). Eight patients had MROs detected at the time of commencement of suppressive therapy, and 2 had MROs detected prior to suppressive therapy (both detected 1 year prior).”

I also want to ask the authors to describe the side effects of the antibiotic therapy in more detail, including gastro-intestinal side effects (e.g. what about nausea, etc.). The side effects mentioned in the table 1: were these the reasons for the cessation of treatment?

Further information about the 2 adverse drug reactions related to cessation of antibiotics has been inserted from line 117:

“One had severe cholestatic liver function derangement (peak GGT 1562, ALP 600) and significant kidney injury (eGFR 37 from 75) with associated metabolic acidosis, which occurred two years following commencement of suppressive therapy. The other patient experienced recurrent vaginal thrush.”

Other side effects not listed in table 1 were identified in the original study conducted on this cohort (reference 3). These were self-reported by patients in a questionnaire conducted in 2015. These have been previously reported and thus not detailed again in this manuscript. The following text has been added to the manuscript at line 123:

“The rest of the reported adverse drug events were noted during our original study (5), and included gastrointestinal side effects such as nausea, vomiting, stomach dis-comfort, reflux and diarrhoea, itch/rash, weight gain, thrush, confusion, alopecia, fatigue, and headache)”

What about the patients who stopped treatment and did not have any recurrence. Can the authors describe these patients in more detail, e.g. what was the time interval for stopping treatment, which diagnoses, was any additional treatment done (like surgery), … Can they give a hypothesis why the cessation of antibiotic treatment was possible? Is there a certain time period that is long enough?

Thank you for these queries, we agree that this information is important to include in the manuscript. An extra column has been added to table 1 to include follow up time. The following details regarding these 6 patients have been inserted into the text at line 80.

“These 6 patients had previously taken prolonged antibiotics for between 2 to 10 years. They were followed up for between 3 months and 2 years following cessation of antibiotics prior to discharge from clinic. Four had prosthetic joint infections and had further surgery (2 had joint revision surgery, 1 had washout/liner exchange, 1 had joint arthroscopy without washout).”

We hope that this revised manuscript is appropriate for publication in Antibiotics.

Reviewer 4 Report

Antibiotics use is inevitable. However, the measurement of the adverse consequences of long-term antibiotic therapy is less. In this study, the authors conducted a 5-year longitudinal study of patients on long-term antibiotics and carried out a retrospective review of medical records and a descriptive analysis. This study is interesting and the manuscript was well-written. The advantage and shortness have bee well discussed. I recommend minor modification before acceptance, like Line 19: microorganisms; Line 19 and 20, bacterial names need to be italic. Also in Table 1 and other places in the manuscript.

Author Response

Dear reviewer,

Thank you for your thoughtful comments and recommendations. We have amended the manuscript to incorporate these changes. Specifically, we have italicised bacterial names and changed the word "organism" to "microorganism" where appropriate. 

We appreciate your review of our manuscript and hope that the revision is now acceptable for publication. 

Yours sincerely,

Jillian Lau

Round 2

Reviewer 3 Report

The manuscript can be accepted in the current form.